


# Radiometric correction of observations from microwave humidity sounders

Isaac Moradi[1,2,3], James Beauchamp[1], and Ralph Ferraro[2]

[1]ESSIC, University of Maryland, College Park, Maryland, USA.
[2]STAR, NOAA, College Park, Maryland, USA.
[3]NASA Global Modelling and Assimilation Office, Greenbelt, Maryland, USA.

*Correspondence to:* Isaac Moradi, ESSIC, University of Maryland, College Park, MD 20740, USA.
(imoradi@umd.edu)

**Abstract.** Advanced Microwave Sounding Unit (AMSU-B) and Microwave Humidity Sounder (MHS) are total power microwave radiometers operating at frequencies near the water vapor absorption line at 183 GHz. The measurements of these instruments are crucial for deriving a variety of climate and hydrological products such as water vapor, precipitation, and ice cloud parameters. However these
measurements are subject to several errors that can be classified into radiometric and geometric errors. The aim of this study is to quantify and correct the radiometric errors in these observations through intercalibration. Since bias in the calibration of microwave instruments changes with scene temperature, a two-point intercalibration correction scheme was developed based on averages of measurements over the tropical oceans and night-time polar regions. The intercalibration coeffi-
cients were calculated on a monthly basis using measurements averaged over each specified region and each orbit, then interpolated to estimate the daily coefficients. Since AMSU-B and MHS channels operate at different frequencies and polarizations, the measurements from the two instruments were not interalibrated. Because of the negligible diurnal cycle of both temperature and humidity fields over the tropical oceans, the satellites with most stable time series of brightness temperatures
over the tropical oceans (NOAA-17 for AMSU-B and NOAA-18 for MHS) were selected as the reference satellites and other similar instruments were intercalibrated with respect to the reference instrument. The results show that Channels 1, 3, 4, and 5 of AMSU-B onboard NOAA-16 and Channels 1 and 4 of AMSU-B onboard NOAA-15 show a large drift over the period of operation. The MHS measurements from instruments onboard NOAA-18, NOAA-19, and MetOp-A are generally
consistent with each other. Because of the lack of reference measurements, radiometric correction of microwave instruments remain a challenge as the intercalibration of these instruments largely depend on the stability of the reference instrument.



## 1 Introduction

Measurements from microwave instruments onboard spaceborne platforms operating near the water

vapor absorption line at $183\,\mathrm{GHz}$ are one of the main sources of observations for tropospheric water vapor, total precipitable water vapor, and cloud ice water path (Ferraro et al., 2005). These data are also increasingly assimilated into NWP models for the purpose of improving weather forecasting or atmospheric reanalyses (Rienecker et al., 2011). AMSU-B and MHS are two of the main microwave humidity sounders that have been flying on NOAA and MetOp satellites since 1998. However, the

measurements of these instruments are subject to several errors that can be classified into radiometric and geometric. Geometric errors are related to a shift in the earth location of measurements and are introduced by sources such as timing error, instrument mounting errors, and errors in instrument modelling and geolocation algorithms (Moradi et al., 2013a). Moradi et al. (2013a) investigated the geolocation errors in these instruments using the difference between ascending and descending ob-

servations along the coastlines and reported several errors including more than one degree antenna pointing error in AMSU-A onboard NOAA-15, about one degree pointing error in AMSU-A2 onboard NOAA-18, as well as a timing error up to 500 milliseconds in NOAA-17. Moradi et al. (2013a) reported generally a relatively good accuracy for the geolocation of AMSU-B and MHS instruments. However, the radiometric errors in these instruments have not yet been fully investigated or corrected

due to the lack of reference measurements.

Once the satellites are launched, it is very difficult to determine the cause of the radiometric errors, but some of the factors that may contribute to these errors include: error in the hot and cold calibration targets, antenna emissivity, Radio Frequency Interference (RFI), antenna pattern correction, and non-linearity in the calibration (Wilheit, 2013; Ruf, 2000; Mo, 2007; Hewison and Saunders,

1996; Chander et al., 2013). The radiometric accuracy of microwave measurements cannot be easily evaluated because of the lack of reference measurements. One main feature of radiometric errors is that the errors are normally scene dependent and change with the scene brightness temperatures and polarization. Over the years some alternative methods have been developed to determine the relative accuracy of microwave measurements, including validation using measurements from sim-

ilar instruments onboard airborne platforms (Moradi et al., 2015a; Wilheit, 2013; John et al., 2012, e.g.,), comparison with simulations conducted using a radiative transfer model and atmospheric profiles (Saunders et al., 2013; Kerola, 2006; Moradi et al., 2013b), and inter-comparison with respect to similar instruments onboard spaceborne platforms (Sapiano et al., 2013; John et al., 2012). Although, the validation versus simulated brightness temperatures can to some extent reveal errors in

microwave satellite measurements, the application is very limited due to the biases in NWP fields, radiosonde sensor biases, as well as errors in the RT models and inputs provided to the RT models such as surface emissivity. One of the methods that has been extensively used to validate the radiometric accuracy of microwave measurements is intercalibration or inter-comparison of data from similar instruments operating on different platforms. In this case, one of the instruments that



is more stable in time is chosen as the reference instrument and all other similar instruments are intercalibrated with respect to the reference instrument. Although, intercalibration cannot be used for absolute validation of microwave measurements, once the reference instrument is determined, other instruments can be relatively validated with respect to the reference instrument. Assuming that data from the reference instrument are stable and valid over time, the intercalibration can serve as a

reliable method to develop homogenized data records from microwave measurements.

     Berg et al. (2016) investigated the radiometric difference between microwave radiometers in the Global Precipitation Measurement Mission (GPM) constellation and reported about 2 K to 3 K difference between most instruments and GPM Microwave Imager (GMI). However, they reported 7 K to 11 K difference between GPM GMI and some of the SSMI channels on board DMSP F19.

John et al. (2012) used global Simultaneous Nadir Observations (SNOs) to intercalibrate microwave humidity sounders (MHS and AMSU-B). Global SNOs normally become available due to orbital drift when the equatorial crossing times of the polar orbiting satellites become close. Based on time/distance match-ups, they suggested a collocation criteria of 5km and 300s for intercalibrating microwave sounders and reported the instrument noise as the major factor affecting the intersatellite

differences. However it should be noted that global SNOs are only available for a limited time-frame and cannot be used to intercalibrate time-series of satellite measurements as the intersatellite differences are expected to vary with time as shown in this paper. Sapiano et al. (2013) used several techniques including polar SNOs, and differences against radiances simulated using a RT model and reanalysis fields, for developing a fundamental climate data records from the Special Sensor Mi-

crowave Imager (SSM/I) radiances. They reported a good agreement between different techniques with a bias of 0.5 K at the cold end and slightly larger bias at the warm end. They reported a smaller intercalibration difference for recent SSM/I instruments (F14 and F15 compared to F13) than for the older instruments (F08, F10, and F11 compared to F13). Saunders et al. (2013) used double difference between brightness temperatures simulated using a RT model and NWP fields and mea-

surements from several MW and IR instruments and concluded that the biases due to NWP models or RT calculations are canceled out by double differences. However, it should be noted that a bias in NWP fields with a diurnal cycle will not be canceled out by double difference techniques as different satellites pass the same regions at different times of the day. Zou and Wang (2011) used global ocean mean differences along with SNOs to intercalibrate radiances of AMSU-A instruments

onboard NOAA-15 to NOAA-18 and MetOp-A. They reported five different sources of biases for intersatellite difference including instrument temperature variability due to solar heating, inaccuracy in the calibration non-linearity, and channel frequency shift. Wessel et al. (2008) used simulated radiances from synoptic radiosondes and NWP models to investigate the calibration of SSMI/S lower atmospheric sounding channels. They reported two major sources of biases including the emissiv-

ity of primary reflector and uncompensated solar heating for the hot load of calibration. Cao et al. (2004) used the Simplified General Perturbation No. 4 (SGP4) to predict SNOs among polar orbiting



satellites. SNO is the most common technique to investigate the intersatellite differences when the two satellites pass over the same region at the same time. A 30-year long fundamental climate data record from HIRS channel 12 clear-sky radiances was produced by Shi and Bates (2011). Shi and
Bates (2011) reported scan-dependent biases causing major differences among the instruments.

The purpose of this research was to quantify and correct the radiometric errors in AMSU-B and MHS observations through intercalibration in order to develop a homogenized data record that can be used for retrieving geophysical variables such as rain rate and tropospheric humidity as well as NWP reanalysis. The rest of this paper is organized as follows: Section 2 introduces the instruments,
Section 3 describes the methodology, Section 4 reports the results, and Section 5 sums up the study.

## 2 Satellite Instruments

AMSU-B and MHS are total power microwave radiometer with 5 channels operating at frequencies ranging from 89 to 190 GHz. AMSU-B was onboard NOAA-15 to NOAA-17 and beginning with NOAA-18, AMSU-B was replaced by MHS. The primary goal of these instruments was for mea-
suring the atmospheric water vapor profiles, but the measurements especially from 89 GHz can also provide information on surface temperature and emissivity (in conjunction with AMSU-A channels) and detect clouds and precipitation. Both instruments have 5 channels, three of which are centered around the water vapor absorption line at 183 GHz. The fourth channel is sensitive to water vapor in moist conditions and surface in dry conditions. The fifth channel is a window channel operating at
89 GHz. The combination of these channels can be used to derive a wide range of atmospheric and hydrological parameters.

All AMSU-B channels and the first channel of MHS are horizontally polarized at nadir, but the third and fourth channels of MHS are vertically polarized. The beam width of AMSU-B is 1.1 degrees but that of MHS is 10/9 degrees. Both instruments are continuous scanners meaning that the
integration is performed while the scanner is moving therefore the effective field of view is larger than instantaneous FOV. The instruments take 8/3 seconds to complete one full scan which includes earth measurements, as well as scanning hot and cold loads. Spatial resolution at nadir is nominally 16 km and the antenna provides a cross-track scan, scanning $\pm 48.95°$ from nadir with a total of 90 Earth FOVs per scan line.

## 125 3 intercalibration method

The most common method for the intercalibration of satellite measurements is to directly compare coincident observations of similar channels on the reference and target instruments. In addition to being measured at the same time and location, these coincident observations should also be measured using the same geometry especially in terms of the earth incidence angle. These coincident
observations are often limited to (near) nadir field of views and are known as Simultaneous Nadir





Observations (SNO). In the case of intercalibrating instruments onboard polar orbiting satellites such as NOAA and MetOp, the SNOs normally occur near the polar region. The differences between reference and target satellites are normally scene dependent, therefore the coincident observations are required to cover a wide range of brightness temperatures. The biggest limitation for finding global

SNOs is that polar-orbiting satellites overpass the same location at different local times. The coincident time requirement for SNOs is because of the diurnal cycle of environmental variables such as temperature, water vapor, clouds and other parameters that affect the satellite radiances. The time requirement can be neglected over regions where diurnal cycle is negligible. There are regions where the diurnal cycle is mainly introduced by random processes and is canceled out after averaging.

For instance, the tropical oceans and polar region during the winter nights are two of these regions where diurnal cycle of the environmental variables such as temperature and humidity impacting the microwave satellite brightness temperatures are negligible. For example, Moradi et al. (2016) reported a negligible diurnal cycle for relative humidity in all layers of the troposphere over the tropical oceans but somewhat significant diurnal cycle over the tropical lands and (Moradi et al., 2015a)

shows that in tropical region, the impact of one hour difference in overpass times on the differences between collocated observations is less than $0.5\,\mathrm{K}$.

Therefore, we employed area averaged brightness temperatures from tropical oceans (tropical band expanding from 30S to 30N) as one intercalibration point and also area averaged brightness temperatures from Antarctica ($<$ 75 S) and Arctic ($>$ 75 N) as the second point of calibration.

There is a small diurnal cycle of temperature and humidity over convective regions of tropical band, therefore we used a cloud filter which is based on the difference between brightness temperatures from different channels to filter out cloud contaminated observations, see Section 4.1. Besides, since in tropical region the diurnal cycles over land can be significant, we only used the area averaged data over ocean. The intercalibration method can be summarized as follows:

– calculate the area averaged Tb's over clear sky tropical oceans and polar nights separately for each instrument and each orbit

    – determine the reference instrument by analyzing the time series of tropical averages as the time series is expected to be stable over time

    – determine the linear relation between area-averaged Tb's for reference and target instruments
160        in a monthly basis

    – interpolate the regression coefficients using cubic-splines to daily values

    – correct the observations from target instrument using the intercalibration coefficients



## 4 Results and Discussion

### 4.1 Cloud Filter

Clouds are expected to have a diurnal cycle especially over the convective regions of tropics, therefore it is required to eliminate convective regions from the intercalibration process. Cloud contaminated observations were filtered using a channel difference as discussed in previous studies (e.g., Moradi et al., 2015b; Buehler et al., 2007). The idea is that because of the lapse-rate in atmospheric temperature, the channels peaking lower in the troposphere have higher brightness temperature than

the channels peaking higher. Therefore, in clear-sky conditions the Tbs of lower channels are warmer than the Tbs of channels peaking higher in the atmosphere. In the case of clouds, the relation is changed as the channels peaking lower are normally more affected by clouds than the channels peaking higher in the atmosphere. Therefore, the channel differences can be used as a filter to remove cloud contaminated observations.

It was found that because of the dry atmosphere in the polar region, the brightness temperatures from channels used to define the cloud filter become sensitive to the surface and the difference between them is not necessarily a function of the cloud optical thickness anymore. Additionally, microwave observations are sensitive to deep convective clouds which are not normally present in the polar region. Therefore, we only applied the cloud filter to observations from the tropical region.

Although, any combination of the differences between channels 3, 4 ,and 5 can be used for the cloud filter, we used the difference between channels 3 and 4 as explained in (Moradi et al., 2015a).

Figure 1 shows a time series of the difference of the differences ($\Delta$) (known as double difference) between clear-sky and all-sky AMSU-B measurements onboard NOAA-15 and NOAA-17 ($\Delta_{clear} - \Delta_{all-sky}$). These double differences show the impact of clouds on the inter-satellite differences. As

shown channel 1 operating at 89 GHz is the most sensitive channel to cloud screening because its Jacobians peak in lower troposphere near the surface, while other channels, in the moist conditions of tropical region, peak in middle and upper troposphere and are less sensitive to clouds.

### 4.2 Diurnal Cycle Effect

The effect of land and ocean on the intercalibration, which is due to a stronger diurnal cycle over

land especially for the near surface-peaking channels, was investigated by separating land and ocean brightness temperatures over the tropical region, then calculating the intercalibration coefficients. Similar to the impact of clouds, we employed double differences to evaluate the impact of larger diurnal cycle over land on the intersatellite differences. In this case, the double difference is calculated as the difference of the differences between land and ocean brightness temperatures of AMSU-B

onboard NOAA-15 versus NOAA-17. If we indicate reference (NOAA-17) and target (NOAA-15) instruments using $r$ and $t$ indices and land and ocean using $L$ and $O$, then the double difference is calculated as $(Tb_{tL} - Tb_{rL}) - (Tb_{tO} - Tb_{rO})$. Figure 2 shows an example of double differences





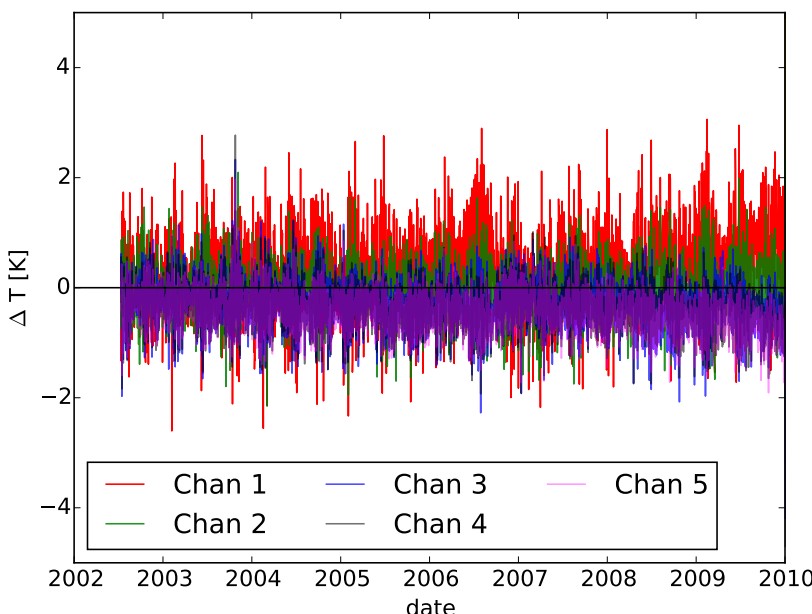

**Figure 1.** Difference between the clear-sky observations of NOAA-15 minus NOAA-17 versus the all-sky observations.

between collocated brightness temperatures of NOAA-17 (reference satellite) and NOAA-15 (target satellite) over land and ocean. As expected, the surface channels are more sensitive to the diurnal

cycle of Tb over land and a small trend is observed that can be explained by the orbital drift of both satellites. The double difference is maximum around 2005 when NOAA-15 ascending (descending) overpass was around 18:00 LT (06:00 LT) and NOAA-17 ascending (descending) overpass time was around 22:30 LT (10:30 LT). Therefore, the intercalibration was limited to tropical oceans to avoid the effect of diurnal cycle. Since during polar winters, that region is normally covered by ice

and snow, we averaged all the data over polar regions and no land/ocean mask was applied. All the experiments for this section were conducted using clear-sky data.

### 4.3 Polarization Difference

Although, AMSU-B and MHS are two similar instruments, there are several differences in terms of polarization and frequency of some of their channels. Both instruments have single polarization at

nadir. All AMSU-B channels and channels 1, 2 and 5 of MHS are vertically polarized but channels 3 and 4 of MHS are horizontally polarized at nadir (Kidwell et al., 2009). The vertical and horizontal


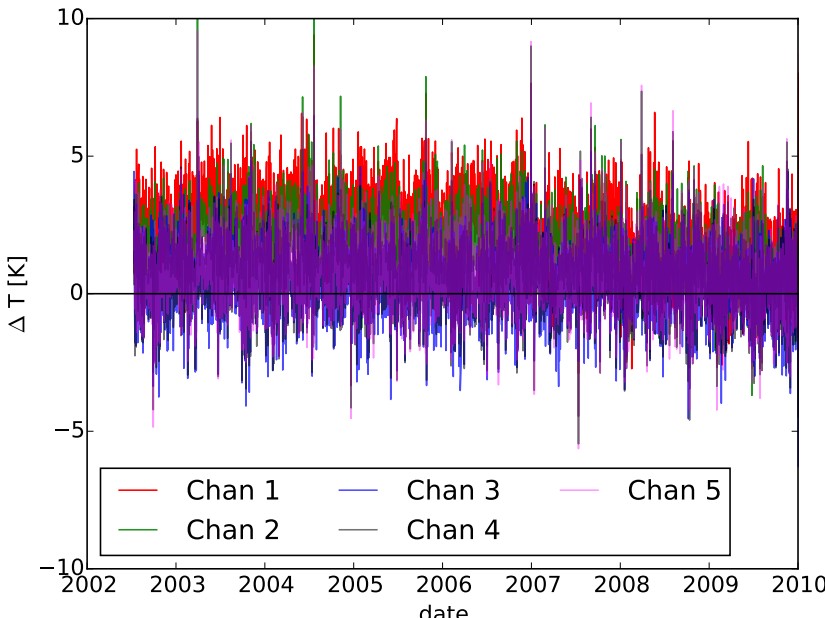

**Figure 2.** Difference between N17 minus N15 over land and ocean averaged over the tropical region.

components of the polarized radiation are the same over ocean at the nadir location, but the polarization changes as the antenna moves toward the edge of the swath. Therefore, the inter-satellite differences at nadir should not significantly depend on the channels' polarization, but as the antenna

rotates the polarization becomes mixed and introduces differences. Other factors that may impact off-nadir differences include the scan-angle dependent bias as well as change in the height of the weighting functions.

Figure 3 shows the inter-satellite differences for NOAA-17 AMSU-B and NOAA-18 MHS for different FOVs. The FOVs' numbers start from the left side of the scan (FOV1), so that the nadir

view is FOV45 and the most right view is FOV90. Note that NOAA-18 overpass time is around 13:00 LT but NOAA-17 overpass time is around 22:00 LT. Therefore, we expect larger differences between the two instruments for the surface channels. This is the case for the sub-nadir observations (FOV45) but for the edge of the scans (FOV01 and FOV90) all the channels behave the same (also see Figure 5). Note that this exercise is not able to rule out other factors that may affect the inter-

satellite differences. One possible explanation is that the weighting functions peak higher as the field of view moves from nadir to the edge of the scan so that some of the FOVs peak high enough in the atmosphere to become insensitive to the surface conditions. However as shown in Figure 4 the



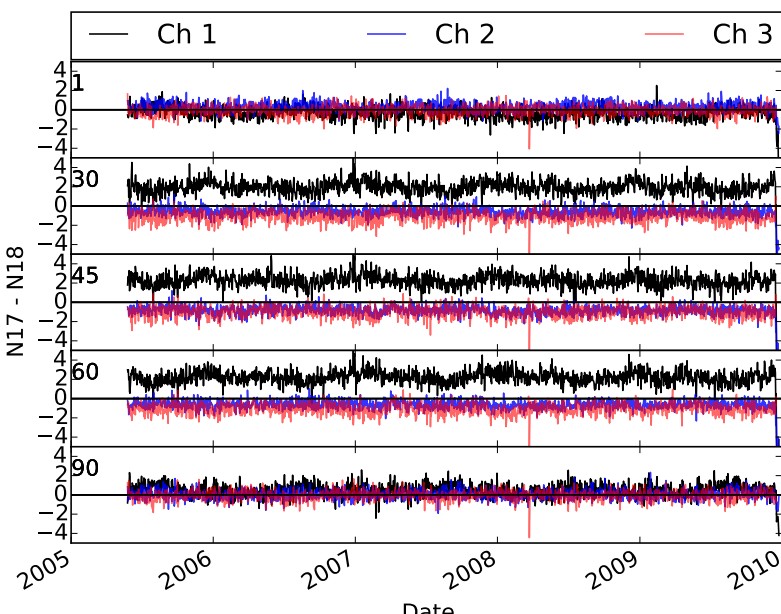

**Figure 3.** Effect of polarization on the difference between NOAA-17 AMSU-B and NOAA-18 MHS observations for different FOVs. FOV numbers are printed on the left side of each suplot.

difference does not seem to be due to difference in over pass time because the differences between the two satellites over land does not show larger differences, though the diurnal variation should be larger over land.

### 4.4 Reference Instrument

As stated before, due to the lack of reference measurements, one of the instruments which is stable over time is chosen as the reference instrument and the other instruments (target instruments) are calibrated with respect to it. Determining the reference instrument is likely to be the biggest challenge in conducting intercalibration. All other instruments will be corrected with respect to the reference instrument, therefore selecting a biased instrument as the reference instrument means that the intercalibrated measurements will suffer from even a larger bias than the original measurements. Because of the lack of reference measurements, it is almost impossible to select an instrument as reference instrument without any uncertainty. One important feature of the intercalibrated measurements is that they are expected to be representative for the climate, thus they may be used for studies related to climate change and variability. As stated before because of negligible diurnal cycle over



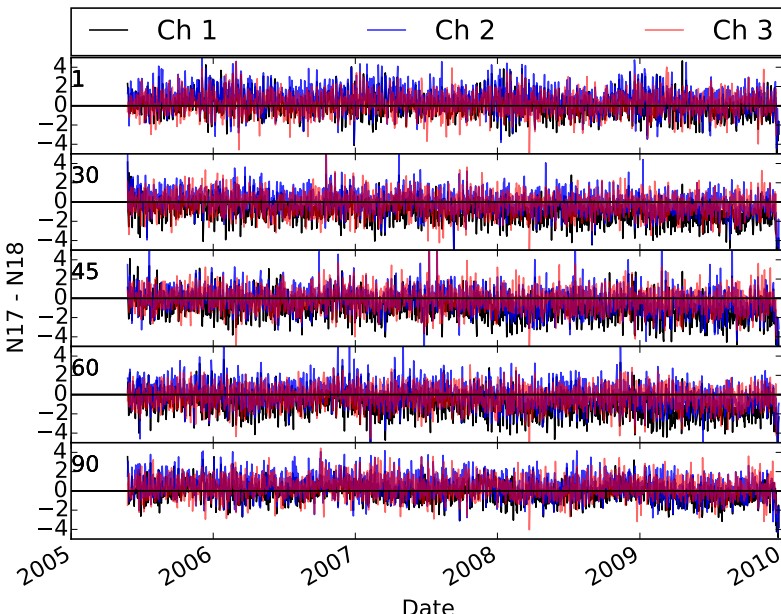

**Figure 4.** Effect of polarization on the difference between NOAA-17 AMSU-B and NOAA-18 MHS observations for different FOVs.

the tropical region, the orbital drift should not introduce a significant trend in the observations. Thus variability in the measurements averaged over the tropical band is expected to be similar to that reported for geophysical variables affecting the brightness temperatures. For instance, variability in

the measurements of surface sensitive channels is expected to be very close to change in surface temperature as the brightness temperatures for those channels are mostly affected by the surface temperature and emissivity. Since the emissivity is not expected to change with time, the variability in the brightness temperatures is expected to follow the change in surface temperature. Figure 6 shows the tropical averages for different satellites and all the five AMSU-B/MHS channels. As

mentioned before we decided not to intercalibrate AMSU-B with MHS measurements, therefore we were required to select one satellite as the reference for the AMSU-B instruments and one for the MHS instruments. NOAA-16 Channels 3-5 show a large drift with time, therefore NOAA-16 was excluded. NOAA-15 experienced some calibration issues especially with regard to RFI, thus we decided to use NOAA-17 AMSU-B as the reference instrument for the AMSU-B instruments. There

is a good consistency between NOAA-17 and NOAA-15 Channel 1 but a systematic difference between AMSU-B and MHS observations for channel 1. Additionally, there is a systematic difference between NOAA-17 Channel 4 and MHS observations for the same channel. Although, NOAA-15



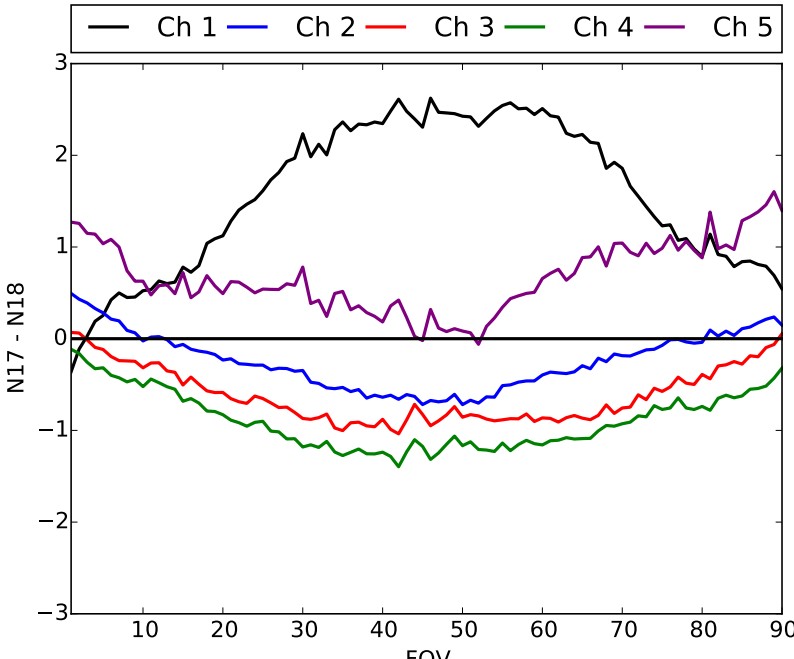

**Figure 5.** Effect of polarization on the difference between NOAA-17 AMSU-B and NOAA-18 MHS observations for different FOVs.

matches with MHS data during that time frame, that is basically caused by the upside and then reverse trend in NOAA-15 observations. The MHS instruments are generally consistent with each other, but we choose NOAA-18 for the reference satellite because the data are available for a longer time period.

### 4.5 Intercalibration Coefficients

The primary measurement of the microwave instruments are digital counts which are converted through a two-point calibration into radiances or brightness temperatures. The calibration equation is based on the relation between digital counts and measured radiances for a radiometrically cold reference (normally when the instrument measures the background space radiance) and a hot (warm) references (normally a blackbody onboard the satellite). The radiometric error can change with the scene temperature if the error is not stable from one load to the other one due to, for instance, non-linearity in the calibration. Because of this scene dependency, it is required to evaluate the inter-satellite differences for a wide range of brightness temperatures. This is one of the main reasons that SNOs are not sufficient for the intercalibration of microwave instruments as SNOs normally



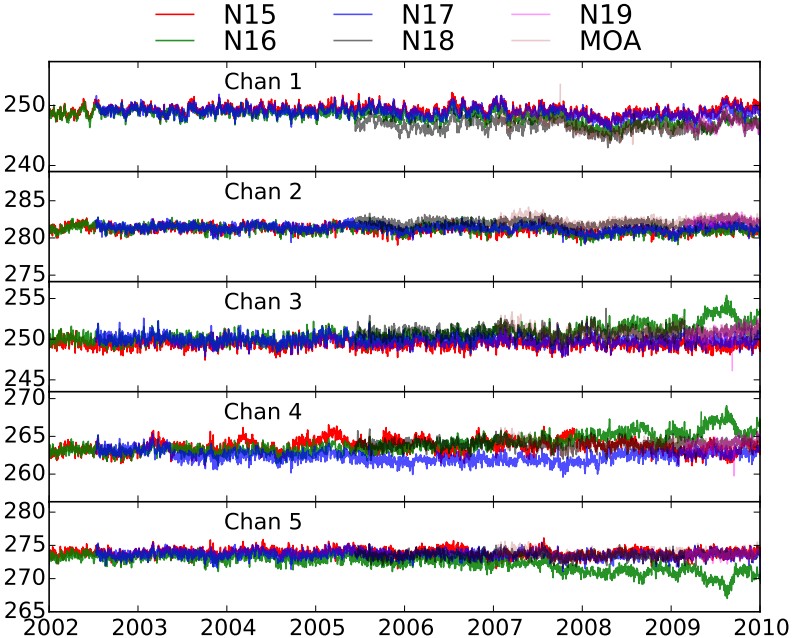

**Figure 6.** Analyzing the time series of observations over the tropical and polar regions for selecting the reference satellites.

occur at high latitudes and only cover a small range of Tbs. In this study, we utilized the averages of brightness temperatures over the tropical region at one end of measurements and polar averages at the other end. Note that either of these can form the lowest or highest values depending on the channel as well as the surface type. As mentioned earlier we only used the brightness temperatures over ocean to calculate the tropical averages.

Figure 7 shows an example of the relation between Tb's from reference and target instruments. Although, zonal averages for mid-latitude oceans are also shown in Figure 7, we only used the polar and tropical averages to calculate the regressions coefficients. All the coefficients are derived using a linear relation as we did not have any evidence of the non-linearity between the differences of target and reference instruments. The calibration coefficients were calculated as $TB_{TARGET} = a x TB_{REFERENCE} + b$. The intercalibration coefficients were calculated in a monthly basis then were interpolated to daily values using spline functions. This helps to reduce the noise in the coefficients. Therefore, the intercalibration process can be explained as follows: (1) data are averaged over the clear-sky tropical oceans and polar nights, (2) one month of data from both regions are used to make the scatter-plots between reference and target satellites, (3) monthly intercalibration





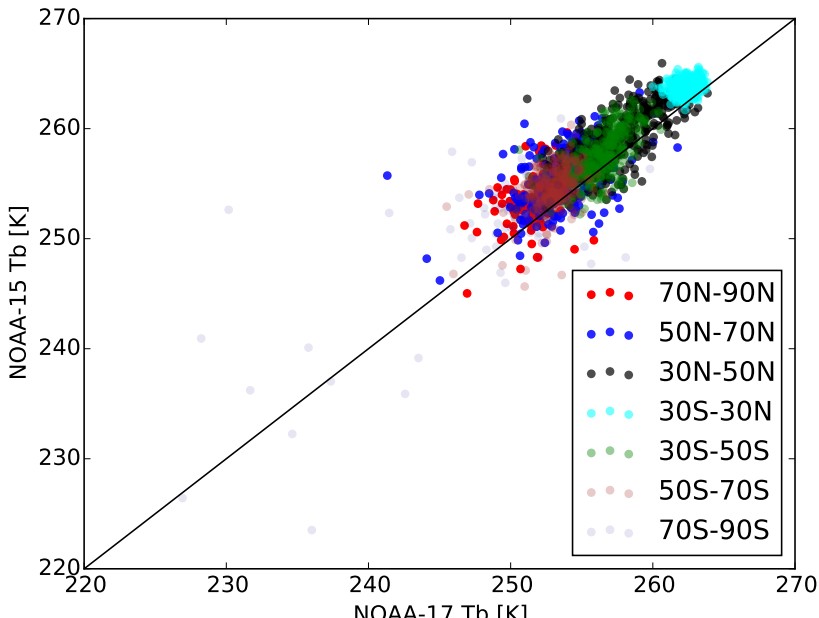

**Figure 7.** Determining the empirical calibration coefficients.

coefficients are calculated then interpolated to daily values, (4) the coefficients are applied to level-1b data to calculate the intercalibrated brightness temperatures.

We did not find any advantage to use moving window averages, i.e., collocate one month of data around the day of interest then move the window to other days. Figure 8 shows an example of monthly values as well as interpolated values. We also found that calculating the intercalibration coefficients on an annual basis is not enough since there might be short term changes in the data that cannot be accounted for using annual coefficients. NOAA-15 was launched in 1998 but NOAA-17 data are only available since 2002. It is not recommended to extrapolate the coefficients therefore the NOAA15 data before 2002 were intercalibrated with respect to NOAA17 data from 2002. The only issues that this causes is that the trend is removed in the dataset so the trend in NOAA15 data before 2001 is not valid.

The results were evaluated using area averaged values over the tropical oceans. Figure 9 shows the time series of intercalibrated brightness temperatures. The time series for NOAA-17 and NOAA-18 are only subtracted from their own mean values for the entire period. Overall, the intercalibrated Tb's are consistent with each other within about 0.5 K. However, there are some periods where the differences are even larger than 1.0 K. The difference between intercalibrated NOAA-15 and NOAA-





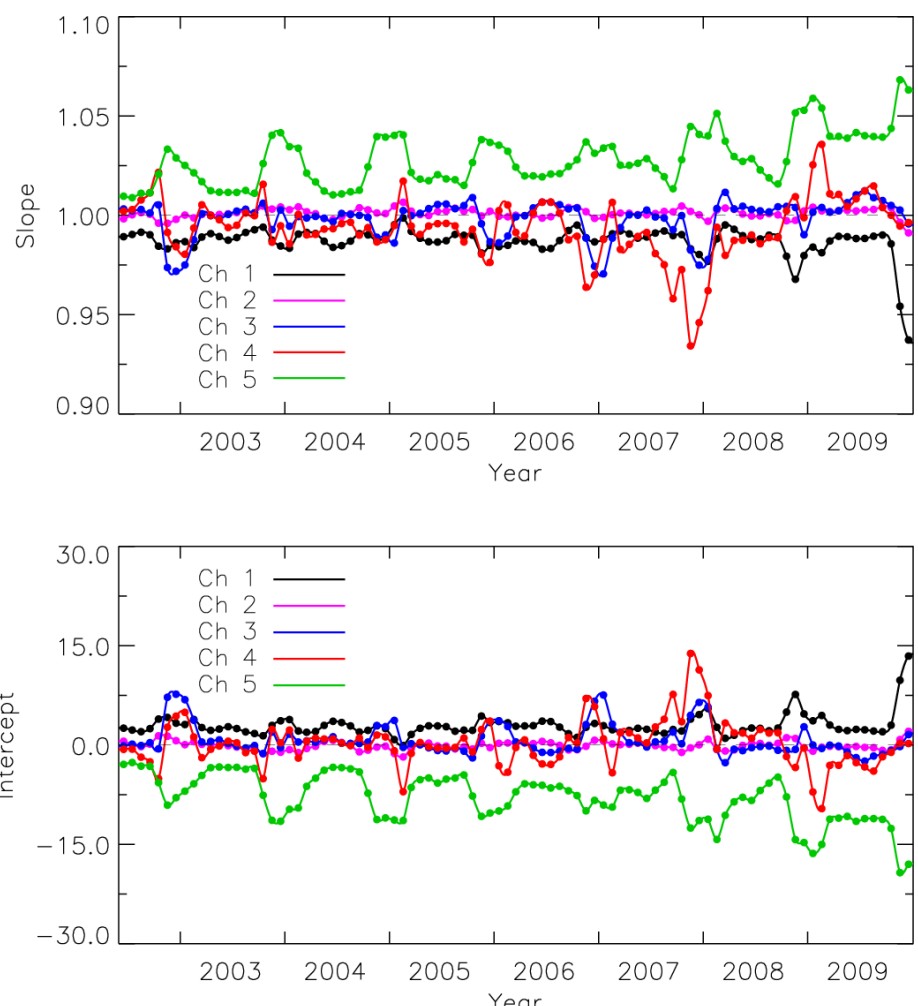

**Figure 8.** Interpolating monthly coefficients using Splines (NOAA-15 vs. NOAA17).

17 observations is generally less than that for NOAA-16 and NOAA-17. For instance, around 2009, NOAA-16 Channel 5 observations show a difference of up to 2 K compared to NOAA-17 Channel 5 measurements. Given that the goal of study was not to completely remove the differences between measurements from different instruments but rather to remove possible biases in the measurements, the consistency observed in Figure 9 is very satisfactory. In the 183 GHz frequencies, one kelvin change in brightness temperature is roughly equal to 7-10% change in relative humidity (Moradi et al., 2015b), therefore it is expected that the derived humidity products have an error less than 10%.





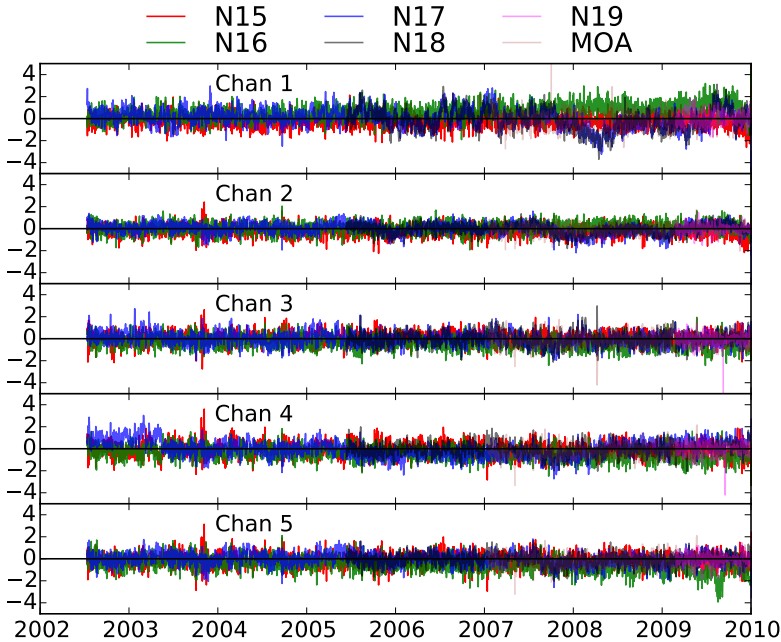

**Figure 9.** intercalibrated time series of AMSU-B and MHS observations.

## 5   Conclusions and Summary

Satellite observations from AMSU-B and MHS are used to retrieve global climate and hyrdrological products such as water vapor, precipitation, and ice cloud parameters. However, these observations are prone to errors and uncertainties that can be classified into radiometric and geometric errors. In the current study, we quantified and corrected the radiometric errors in these observations for the period of 2000-2010. The AMSU-B observations suffer from several instrument failure after 2010, the work is currently under progress to correct some of the AMSU-B observations for the period 2010-2015. A unique characteristic of the radiometric error is that it changes with the scene temperature. A common technique that is used for the radiometric correction is intercalibration of observations measured by similar instruments. A key parameter in intercalibrating satellite observations is to find coincident observations or observations for the same location and same time. Since finding such coincident observations is challenging, we used daily averages of brightness temperatures over regions with negligible diurnal variations. In this study, we used monthly averages of measurements over the tropical oceans and night-time polar regions to perform the intercalibration. In this two-point scheme, the intercalibration coefficients are calculated using monthly averages then interpolated to the daily values using a cubic spline. We selected AMSU-B onboard NOAA-17 as the reference



instrument for all AMSU-B instruments and MHS onboard NOAA-18 as the reference for all MHS instruments. We did not intercalibrate AMSU-B and MHS because of some differences in the frequency and polarization among the two instruments. Most AMSU-B channels onobard NOAA-16

and Channels 1 and 4 of AMSU-B onboard NOAA-15 showed a large drift with time which were corrected with respect to NOAA-17 data. Measurements from MHS instruments were very consistent with each other. Selecting a reference instrument is the most challenging part of the intercalibration because of the lack of reference observations. Selecting a biased reference instrument means that all the intercalibrated measurements will be biased. Another challenge is the intercalibration of

cloud contaminated observations. Due to a larger diurnal variation for the clouds over the tropical regions, we only used clear-sky observations to perform the intercalibrations. Neither the simultaneous nadir observations nor the technique used in this study can be used for the intercalibration of cloud contaminated measurements because of the dynamic nature of the clouds.

*Acknowledgements.* This study was supported by NOAA grant# NA09NES4400006 (Cooperative Institute for

Climate and Satellites - CICS) at the University of Maryland, Earth System Science Interdisciplinary Center (ESSIC). The views, opinions, and findings contained in this report are those of the authors and should not be construed as an official National Oceanic and Atmospheric Administration or U.S. Government position, policy, or decision.



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
