# Peer review of "Radiometric correction of observations from microwave humidity sounders"

_Atmospheric Measurement Techniques, 2018_

## Referee Comment (RC1) · Anonymous Referee #1 · 28 Sep 2018

General Comments

Creating a consistent long-term observation dataset for microwave humidity sensors is an important topic of study and the dataset described in the paper is widely used by many in the scientific community. Users will find this manuscript to be a valuable reference. There are several points of clarification that will improve the manuscript as I have outlined below. I recommend for publication after these are addressed.

Specific Comments

Section 2 should include a listing of the frequencies associated with each channel number. I am assuming Channel 1 is 89 GHz, Channel 2 is 150 GHz etc., but this is not stated. Also, it is confusing that you say "fifth channel...89 GHz" when you do in a

later section say that Channel 1 is 89 GHz. I would not use the labels of "first, second, etc." unless they directly correspond to Channels 1, 2, etc.

Section 2, second paragraph, first sentence. AMSU-B is vertically polarized, while Channels 3 and 4 of MHS are horizontally polarized (you have these switched in the sentence).

I would like to see some more details about how the polar regions are used in the intercalibration. What kind of filtering was done for the area averaged brightness temperatures over the Antarctic and Arctic? Is there a reference or some kind of evidence to show that the diurnal cycle of temperature and humidity is negligible in the polar region? Page 5, 1st paragraph mentions how the diurnal cycle in polar regions and tropics are negligible but only gives references to back up this claim for the tropics. Also, in the polar regions, some of the channels especially the window channels see the surface, which will change seasonally. How is surface variability accounted for in these channels so it doesn't impact your intercalibration and cause a seasonal signal?

In Figures 1 and 2 it is really hard to see any trends in the data as its rather noisy and all the channels are plotted on top of each other. Page 7, 1st paragraph refers to some trends that can be seen in Figure 2 but this is a bit hard to see. Perhaps you could do separate subplots for each channel, and maybe plot a running average on top of the raw data so that trends can be more easily observed?

Is Figure 3 the intersatellite differences for the tropics? And it sounds like Figure 4 shows the same thing as Figure 3 but is over land while Figure 3 is over ocean only? Please make this more clear in the text as well as the figure labels.

Figure 5 does not appear to be referenced in the text except on page 8 (where it says "see Figure 5"), but there is no description in the text for what exactly Figure 5 shows. Is it an average over the years for the ocean measurements?

Figure 6 caption says "time series... tropical and polar regions", however from the text

it sounds like this is showing only the tropical regions.

Figure 7: The really light colors (for 50S-70S and 70S-90S) are hard to see. I recommend making the colors darker.

Figure 7: The southern polar region (70S-90S) shows a lot of variability with some extreme outliers. Are all values averaged or some filtering done to remove these outliers?

Figure 8: Channel 5 slope and intercept appear to have a seasonal signal associated with it. Any idea why this is? I would be concerned that a seasonal signal is being incorporated into the intercalibration.

Technical Corrections

Remove commas after the word "Although" when used at the start of a sentence (this happens many times throughout the manuscript).

Page 11, line 267. Change "references" to "reference".
* * *

---

## Short Comment (SC1) · 11 Oct 2018

It was with great interest that I read the manuscript amt-2018-252 and the referee's knowledgeable comments. Radiometric correction of microwave sounders is certainly an important task. One result presented by the authors particularly caught my attention, viz. the failure of the intercalibration coefficients to remove the bias between NOAA-16 and -17 for channel 5 after 2006. As I did not find an explanation for this anomaly in the manuscript, I take the liberty of offering one myself: radio frequency interference (RFI) in combination with a strongly decreasing gain (see amt-11-4005-2018). As RFI has got nothing to do with the scene temperature, it cannot be corrected with the coefficients calculated by the authors.

[Figure]

They list many other possible sources of bias as well, so I suggest to add a discussion of why a linear function of brightness temperature is considered sufficient to deal with all of them.

---

## Referee Comment (RC2) · Anonymous Referee #2 · 22 Oct 2018

[General] This manuscript shows the methods and approaches of intercalibration between sounders to apply for long-term data set like climate data record. However, as I mentioned initially, Authors need to improve manuscript including figures and clarify the description helping readers to understand/read easily. I recommend to fix following issues before publish the paper.

[Misspell] Line 13, interralibrated should be intercalibrated Line 195 and 329, onobard should be onboard Line 312, hyrdrological should be hydrological Line 281-282, recommend target/reference to be lower case Line 282, ax should be a only? (if a and b are slope and intercept) Line 117, horizontally should be vertically Line 118, vertically should be horizontally

[Comment] Line 17-18, it is better to show the table or description of channel, frequency, polarization and use channel number thereafter. Figures 1 and 2 are need to improve to help readers can follow the description. Suggest separate by channel like Figure 6. Figure 3 and 5 are basically telling same story, yet prefer to see Figure 5 since it has all channels and better understanding. Line 198, please specify the temporal/spatial condition for your collocation. Figures 3 and 4 are not included channel 4 and 5. Is it Figures 6 and 9, please specify the MOA as MetOp-A somewhere. I assume NOAA-19 and MetOp-A has been doing same procedure as others using NOAA-18 as a reference instrument. Figure 6, is this both tropical and polar regions or just only tropical region? Figure 7, it's better show only tropic (30S-30N) and polar ($\sim$75S, 75N$\sim$) only and then plot your calibration coefficients line on top of them. (specify a and b as well)

---

## Author Comment (AC1) · 12 Nov 2018

**Reviewer #1**

We sincerely appreciate the reviewer for carefully reading the manuscript and providing feedback. We have considered all the comments in the revised manuscript. A short answer to the comments is also given below.

5 Section 2 should include a listing of the frequencies associated with each channel number. I am assuming Channel 1 is 89 GHz, Channel 2 is 150 GHz etc., but this is not stated. Also, it is confusing that you say "fifth channel...89 GHz" when you do in a later section say that Channel 1 is 89 GHz. I would not use the labels of "first, second, etc." 10 unless they directly correspond to Channels 1, 2, etc.

We revised Section 2 to avoid inconsistency in the text and also included the channel frequencies along with the channel numbers that we have used throughout the text. It now reads as follows:

AMSU-B channels 1-5 operate at 89.0, 150.0, 183.3±1.0, 183.3±3.0, 183.3±7.0 GHz, respectively and MHS Channels 1-5 operate at 89.0, 157.0,  $183.3\pm1.0$ ,  $183.3\pm3.0$ , and 190.3 GHz, respectively. The combination of these channels can be used to derive a wide range of atmospheric and hydrological parameters.

Section 2, second paragraph, first sentence. AMSU-B is vertically 15 polarized, while Channels 3 and 4 of MHS are horizontally polarized (you have these switched in the sentence). We changed this sentence to

AMSU-B channels are all vertically polarized at nadir(Hewison and Saunders, 1996), but MHS Channels 3 and 4 are horizontally and the rest are vertically polarized at nadir (Kidwell et al., 2009).

I would like to see some more details about how the polar regions are 20 used in the intercalibration. What kind of filtering was done for the area averaged brightness temperatures over the Antarctic and Arctic?

We have revised Section 3 to better explain the intercalibration method, especially the filters that we have used in polar regions. Please see the revised manuscript.

Is there a reference or some kind of evidence to show that the diurnal 25 cycle of temperature and humidity is negligible in the polar region? Page 5, 1st paragraph mentions how the diurnal cycle in polar regions and tropics are negligible but only gives references to back up this claim for the tropics. Also, in the polar regions, some of the channels especially the window channels see the surface, which will change sea-30 sonally. How is surface variability accounted for in these channels so

it doesn't impact your intercalibration and cause a seasonal signal? We have included references (e.g., see Przybylak, 2016, Figure 4.3) that show the diurnal variation of temperature in polar region is negligible and more importantly the diurnal variation is not systematic. As mentioned by the reviewer, many of the channels would be significantly impacted by

the surface during polar night because of dry atmosphere. Therefore the factors that would impact 35 the brightness temperatures are surface emissivity and surface/skin temperature. Given that the surface temperature doesn't change much during polar nights and the emissivity remains significantly constant as the surface cover will not change systematically during day, we don't expect any significant or systematic change in measured brightness temperatures over the course of the day. We are

40 collocating the observations in a daily basis so the seasonal variations should not affect our results. We have revised the manuscript accordingly to better discuss these parameters.

In Figures 1 and 2 it is really hard to see any trends in the data as its rather noisy and all the channels are plotted on top of each other. Page 7, 1st paragraph refers to some trends that can be seen in Figure 45 2 but this is a bit hard to see. Perhaps you could do separate subplots

for each channel, and maybe plot a running average on top of the raw data so that trends can be more easily observed?

We have plotted individual channels in separate plots (including weekly moving averages) to be able to better show the change in the window channels. We appreciate the reviewer for carefully reading and checking the text and the plots. Please see the revised manuscript for the new plots.

Is Figure 3 the intersatellite differences for the tropics? And it sounds like Figure 4 shows the same thing as Figure 3 but is over land while Figure 3 is over ocean only? Please make this more clear in the text as well as the figure labels.

55

50

We have amended both the captions of the figures as well as the text to better reflect these differences. Now it reads as follows:

Figure 3 shows the inter-satellite differences for NOAA-17 AMSU-B and NOAA-18 MHS versus FOVs averaged over tropical oceans for the entire period. The FOVs' numbers start from the left side of the scan (FOV1), so that the nadir view is FOV45 and the most right view is FOV90. Note that NOAA-18 overpass time is around 13:00 LT but NOAA-17 overpass time is around 22:00 LT. As shown in Figure 3, the differences between the two instruments significantly change with FOV especially for Channel 1. Figure 4 shows the time series of the differences between the two instruments. As shown in Figure 4, the differences exist for the entire period and other than some small variations, do not vary with time. Figure 5 shows the difference between the two instruments over tropical land. If the differences were due to different overpass times then the differences between the two instruments should be larger over land. However not only are the differences generally smaller over land but also they do not depend on the FOV. Since the ocean is a polarizer in MW frequencies but the land generally is not a polarizer, the difference between Figures 4 and 5 particularly highlights the effect of polarization on the differences between the two instruments over tropical oceans. Note that this exercise is not able to rule out other factors that may affect the inter-satellite differences. One possible explanation is that the weighting functions peak higher as the field of view moves from nadir to the edge of the scan so that some of the FOVs peak high enough in the atmosphere to become insensitive to the surface conditions.

Figure 5 does not appear to be referenced in the text except on page 8
(where it says "see Figure 5"), but there is no description in the text
60 for what exactly Figure 5 shows. Is it an average over the years for the
ocean measurements?

We have amended the text to better reference Figure 5. Please see our answers to the previous comment and note that Figures are rearranged so Figure 5 is now Figure 3.

Figure 6 caption says "time series... tropical and polar regions", 65 however from the text it sounds like this is showing only the tropical regions. We have amended the figure caption to reflect the correct dataset.

```
Figure 7: The really light colors (for 50S-70S and 70S-90S) are hard to see. I recommend making the colors darker.
```

70 We have included a revised version of Figure 7.

Figure 7: The southern polar region (70S-90S) shows a lot of variability with some extreme outliers. Are all values averaged or some filtering done to remove these outliers?

Figure 7 now have all the channels included. Besides we have removed the data that were not used
int he regressions (daytime polar regions as well as mid-latitude ocean averages). The outliers were from the day-time polar regions and don't exist in the new figure.

Figure 8: Channel 5 slope and intercept appear to have a seasonal signal associated with it. Any idea why this is? I would be concerned that a seasonal signal is being incorporated into the intercalibration.

80 There is generally a trade-off between regression coefficients (slope and intercept). In this case as well as Channel 4 early 2008 (for example) the coefficients can be stabilized by putting thresholds on the least square minimization but of course the results would be essentially the same. However we preferred to avoid constraining our regression coefficients and let the minimization fully be automated.

85 Technical Corrections:

Remove commas after the word "Although" when used at the start of a sentence (this happens many times throughout the manuscript). Page 11, line 267. Change "references" to "reference". Done!

---

## Author Comment (AC2) · 12 Nov 2018

**Reviewer #2**

90  [General] This manuscript shows the methods and approaches of inter-calibration between sounders to apply for long-term data set like climate data record. However, as I mentioned initially, Authors need to improve manuscript including figures and clarify the description helping readers to understand/read easily. I recommend to fix following
95  issues before publish the paper.

We thank the reviewer for taking time to carefully read the manuscript and make comments. We have taken all the comments into account in revising the manuscript. A brief answer to individual comments is given below and we have also included the annotated manuscript that shows the revisions.

100  [Misspell] Line 13, interralibrated should be intercalibrated Line 195 and 329, onobard should be onboard Line 312, hyrdrological should be hydrological Line 281-282, recommend target/reference to be lower case Line 282, ax should be a only? (if a and b are slope and intercept) Line 117, horizontally should be vertically Line 118, vertically
105  should be horizontally

All the typos are corrected and other recommendations were taken into account. Additionally we proofread the manuscript again.

[Comment] Line 17-18, it is better to show the table or description of channel, frequency, polarization and use channel number thereafter.
110  We have amended that specific section to better explain the instruments. The text now includes the frequencies for all the channels.

Figures 1 and 2 are need to improve to help readers can follow the description. Suggest separate by channel like Figure 6.

We made these figures again and have separated the plots by channel. Please see the revised
115  manuscript.

Figure 3 and 5 are basically telling same story, yet prefer to see Figure 5 since it has all channels and better understanding.

We agree that there is overlap between Figures 3 and 5, however Figures 3 also shows the time series and possible changes in the differences. So respectfully we would prefer to keep both figures.

120  Line 198, please specify the temporal/spatial condition for your collocation.

The details of collocation criteria are described in Section 3. It is basically based on using area averaged values from tropical oceans and night-time polar regions.

Figures 3 and 4 are not included channel 4 and 5.
125  We didn't include Channels 4&5 because those two channels are very similar to Channel 3. We explained it in the caption to clarify why those channels are not shown

Is it Figures 6 and 9, please specify the MOA as MetOp-A somewhere.
We have included this in the caption of the figures.

4

No the caption of Figure 6 reads as:

> Analyzing the time series of observations averaged over the tropical oceans for selecting the reference satellites. NOAA-19 and MetOp-A (MOA) are intercalibrated with reference to MHS onboard NOAA-18.

130 and the caption of Figure 9 reads as:

> Intercalibrated time series of AMSU-B and MHS observations. NOAA-19 and MetOp-A (MOA) are intercalibrated with reference to MHS onboard NOAA-18.

I assume NOAA-19 and MetOp-A has been doing same procedure as others using NOAA-18 as a reference instrument.

Yes that is correct! It is now described in the caption of the figures too. Please also see our answer to the previous comment.

135 Figure 6, is this both tropical and polar regions or just only tropical region?

Only tropical oceans. We also amended the caption to reflect this fact.

Figure 7, it's better show only tropic (30S-30N) and polar (75S, 75N) only and then plot your calibration coefficients line on top of them.
140 (specify a and b as well)

We made the plots again and limited the data to only tropics and polar nights and have printed the coefficients on the plots.

---

## Author Comment (AC3) · 12 Nov 2018

The comment was uploaded in the form of a supplement:
https://www.atmos-meas-tech-discuss.net/amt-2018-252/amt-2018-252-AC3-supplement.pdf
* * *